# Internal and External Barriers to Bladder Management in Persons with Neurologic Disease Performing Intermittent Catheterization

**DOI:** 10.3390/ijerph20126079

**Published:** 2023-06-08

**Authors:** Amber S. Herbert, Blayne Welk, Christopher S. Elliott

**Affiliations:** 1Department of Urology, School of Medicine, Stanford University, Stanford, CA 94304, USA; 2Department of Surgery and Epidemiology and Biostatistics, Western University, Ontario, ON N6G 2M1, Canada; 3Division of Urology, Santa Clara Valley Medical Center, San Jose, CA 95128, USA

**Keywords:** neurogenic bladder, intermittent catheters, toilet facilities, spinal cord injury

## Abstract

People living with neurogenic lower urinary tract dysfunction (NLUTD) often have to use clean intermittent catheters (CIC) to manage their bladder function. The use of catheters presents multiple unique challenges, based on both the person’s inherent characteristics and on the external limitations imposed by public toilets. We review the impact of age, sex, upper limb function, caregiver assistance, time required to perform CIC, and urinary incontinence on CIC in NLUTD, with special reference to their interaction with societal and public health factors. Public toilet limitations, such as lack of availability, adequate space and special accommodation for CIC, cleanliness, and catheter design are also reviewed. These potential barriers play a significant role in the perception and performance of bladder care in people living with NLUTD.

## 1. Introduction

Neurogenic lower urinary tract dysfunction (NLUTD) is often a sequalae of neurologic disorders such as spinal cord injury (SCI), multiple sclerosis (MS), and spinal bifida (SB) [1,2], with the location and degree of the neurologic impairment usually determining the pattern of lower urinary tract dysfunction. Often, a patient-specific approach is needed to manage a patient’s bladder, with a large majority of those affected requiring the use of either intermittent catheters or indwelling catheters due to a failure to volitionally void (70% of those with SCI, 15% of those with MS, and 85% of those with spina bifida) [3]. In those unable to volitionally void secondary to their neurologic disease, clean intermittent catheterization (CIC), the act of passing a catheter through the urethra to drain the bladder at regular intervals, is often touted by health professionals as the gold standard for bladder management, because of literature demonstrating that it decreases overall morbidity in populations with NLUTD [3].

NLUTD patients who require catheters often report a lower overall reported quality of life [1]. While the ultimate goal in those with NLUTD is to maintain normal overall health and renal function, renal deterioration in this population is now rare, owing to physicians having a proper understanding of urinary tract physiology and proper bladder management. As a result, the “four I’s” have taken on increased importance in the clinical care of these patients: (1) avoiding urinary Incontinence; (2) avoiding urinary tract Infections; (3) achieving Independence; and (4) minimizing Inconvenience [4], all in an effort to promote optimal bladder-specific quality of life [5]. Despite CIC being recommended as the gold-standard for NLUTD management in those unable to volitionally void, it is not always the most feasible option for patients and their families. Consequently, some describe significant negative impacts associated with CIC, and many patients shift to different bladder management options with time [1,6,7]. The objective of this review is to outline some of the significant internal and external barriers that can be encountered by people with NLUTD who perform CIC.

## 2. Internal Barriers

Internal barriers are those that are in the immediate surrounding of the person using CIC or those that are inherent to the person using CIC. Common internal barriers for people living with NLUTD, include age and its interaction with other factors, female sex, hand function, time required to catheterize, urinary incontinence, and limited help from caregivers. These barriers can be daunting to individuals who select CIC as their long-term management option [8]. 

### 2.1. Age

Age alone is not an internal barrier that NLUTD patients face. In younger patients, Castillo et al. found individuals between 2–17 years of age were able to self-catheterize, with most being able to perform CIC completely independently at 9 years old [9]. In individuals, >65 years old, approximately 80% of a predominantly male population of patients were able to successfully learn clean intermittent self-catheterization [10]. Similarly, in an older, predominantly female population with a mean age of 76 years old, more than half were able to successful perform CIC [11]. In addition to sex (discussed further in Section 2.2), risk factors that may preclude elderly individuals from performing CIC include obesity, decreased functional independence, and cognitive disorders such as dementia [10].

### 2.2. Women

In females with NLUTD, the requirement to perform CIC often poses greater challenges than it does for males, secondary to the internal location of the female urethra. Hence, the technique for women to self-catheterize is often more involved [12], and can prevent a woman from catheterizing if she uses a wheelchair. Instead, many women report that to perform CIC they must transfer over to a toilet to have proper urethral access [13]. In even more problematic situations, a woman may not be able to self-catheterize in a sitting position, and instead must transfer to a supine position prior to self-catheterization; depending on the neurologic lesion and any functional limitations in their legs and trunk, this may require an attendant or lift. In either situation, bladder management in a public setting is almost impossible, and the woman’s ability to leave the home for long periods of time is limited.

In overweight and obese women, CIC is often even more challenging due to increased difficulty accessing the urethra, with data suggesting that CIC is less likely to be adopted than in normal weight women [14]. Lower extremity spasticity (which is common with SCI or MS) can also make positioning and accessing the urethra more difficult in the female NLUTD population. These barriers, especially when more than one is present, can make CIC an anxiety-provoking experience for women (to a much higher degree than their male counterparts), with some perceiving self-catheterization as a traumatic experience and interpreting it as being disturbing [12].

### 2.3. Upper Extremity (UE) Motor Function

Intermittent catheterization is often also challenging for individuals whose upper extremity motor function prohibits them from removing pieces of clothing, holding a catheter, or accessing their genitalia [12]. As CIC generally requires two working upper extremities to remove clothing and provide access to the urethra, many with neurologic disease can struggle with the steps of CIC. In the SCI population, over 40% have less than optimal UE motor function for performing CIC [15], with many valuing any modality that could help them regain partial hand and arm function [16]. In those with less than adequate upper extremity motor function, choosing CIC often requires a reliance on caregivers to assist with bladder management. The inconvenience and lack of independence associated with having an around-the-clock caregiver for their bladder leads to over half of the SCI population reverting to indwelling catheterization long-term, with those with impaired UE motor function being most likely to discontinue CIC [17].

### 2.4. Lack of Caregiver Assistance

Even in situations wherein someone with NLUTD might consider utilizing caregiver assistance to aid in their bladder management, caregiver availability can be limited, as insurance coverage is often lacking. With the cost of hiring paid provider services in the United States approaching $28,000/year on average [18], many with NLUTD are unable to afford this out-of-pocket expense. As a result, family members who are already faced with countless other issues related to caring for a person with physical disabilities, are often the only option for someone with NLUTD needing to perform scheduled CIC both at home and in public [19]. In these situations, it has been shown that adherence to a catheterizing schedule is often not rigorous, possibly because patients who are reliant on their caregivers for catheterization may be reluctant to increase the number of catheterizations per day out of fear of increasing caregiver burden [8]. These issues are often compounded by caregiver resistance to learning and performing CIC [20], as the burden of care for a non-professional can become overwhelming.

### 2.5. Time to Perform CIC

In addition to anatomic or UE motor limitation difficulties, there is also an increased time burden required to perform clean intermittent catheterization (53.4 min per day) compared to using an indwelling catheter (17.4 min per day) [21]. In [8] While surgical reconstruction in the form of a catheterizable stoma can decrease the time taken to catheterize (to approximately 8 min) [22] in select patient groups, many patients are unwilling to undergo the necessary invasive procedures, especially in light of the 30% long-term reoperation rate (even when performed by expert surgeons) [23].

### 2.6. Urinary Incontinence

Urinary incontinence in patients with NLUTD is common (experienced by approximately 50% of people with MS and SCI) and decreases overall quality of life [24,25]. Frequent episodes of urinary incontinence can preclude someone with NLUTD from going out in public and participating in their daily activities of life. In hopes of minimizing urinary incontinence, individuals will often decrease fluid intake and self-catheterize before they go to leave their home environment [26]; however, this is not always effective. If a patient has an episode of urinary incontinence in public, they face the social stigma of smelling of urine and may feel the need to change their clothes, which is an additional challenge, especially when coupled with their motor limitations. In all, urinary incontinence can lead to embarrassment, social isolation, and a worse quality of life [27].

## 3. External Barriers

Aside from the internal barriers discussed above, there are several external barriers someone with NLUTD may face that can prevent adoption and adherence of CIC. External barriers are those that present in specific situations, usually when outside of the home. In an attempt to mitigate external barriers to disabled persons, The Americans with Disabilities Act (ADA) was created in 1990 to provide protections to those with disabilities in all areas of life including transportation, employment, and private and public spaces that are open to the general public [28]. While well-intentioned, these laws and improvements do not always meet the needs of the NLTUD population, for a variety of reasons that are discussed below.

### 3.1. Appropriate Space/Lack of Accessibility

Despite governed civil rights law, individuals with NLUTD continue to face barriers to bladder management in public spaces. One of these barriers is the availability of wheelchair-accessible stalls [13], as smaller bathroom stalls that provide limited space for proper positioning and even less room for those who require caregiver assistance are not good for performing CIC. In addition, CIC often requires multiple supplies positioned near the toilet (lubricant, urine drainage receptacle, etc.) and clean shelves or tables on which to unpack and place the necessary equipment; however, these are often not available in public restrooms. Likewise, public travel is often difficult for those with NLUTD due to a lack of space in airplane, train, or bus bathrooms. In many cases, individuals performing CIC will place an indwelling catheter prior to travelling to avoid this problem, or may turn to indwelling catheters long-term to avoid these public facility limitations [29].

### 3.2. Lack of Facilities

ADA Standards for bathrooms require that at least 50% of toilets in single, clustered toilet rooms be handicap-accessible [28]. Many public facilities are only partially compliant with ADA mandates on restrooms [30]; they often have only one large wheelchair accessible stall, despite having more than two other toilet stalls [13]. The lack of facilities may require individuals to self-catheterize in less-than-ideal scenarios, such as in their vehicle, and may further limit their ability to spend time in public. The lack of adequate restrooms extends to the private homes of close families and friends [31]. Some patients have improvised by taking small urinals to their friends’ homes to self-catheterize; nonetheless, this can be demoralizing for individuals with NLUTD [31].

### 3.3. Spaces in Use by Those That Are Not Handicapped

Even when bathrooms have a wheelchair-accessible stall, an NLUTD patient can find the stall occupied by individuals who are not disabled, forcing them to wait for prolonged time periods to drain their urine. This has the potential to lead to a higher risk of UTIs, incontinence episodes, and autonomic dysreflexia in vulnerable individuals [13]. In urgent urinary scenarios, those with NLUTD may be forced to undress in the main public area of the bathroom prior to being transferred into a smaller, non-wheelchair accessible stall, with the assistance of a caregiver [13]. This loss of privacy and independence can be troubling and may further reinforce the idea that public participation should be avoided, or that CIC is associated with significant independence limitations.

### 3.4. Lack of Cleanliness in Public Restrooms

While lack of public restroom cleanliness is often off-putting to all individuals, in the NLUTD population, who are already more prone to UTIs than the general population, the situation can be even more unnerving [32]. Typical catheterizing protocols, which are often modified even in the best of situations, can be compromised further in unclean facilities, again further reinforcing one’s propensity to avoid public participation.

### 3.5. Lack of Public Understanding

Another barrier to performing CIC is that the general public lacks insight and often has negative perceived attitudes of those with NLUTD [31]. A large proportion of the general public is unaware that many with NLUTD are unable to volitionally void and are instead reliant on catheters to empty their bladder [13]. As a consequence, many in the general public do not understand the coordinated process that is required for individuals to perform CIC in a public setting, and might not understand how their use of a handicap stall might adversely affect someone with a neurologic disability, when other appropriate facilities are not present [13].

### 3.6. Type and Cost of Intermittent Catheters

Since CIC has become the gold standard for bladder management in NLUTD, there has been an increase in the variety of catheters available to patients. An individual may select one catheter based on various considerations, including streamlined packaging, ease of grip, catheter length, and anatomic considerations (such as a coude tip for urethral false passages or enlarged prostates) [33]. Self-lubricating and water-based or hydrophilic catheters have become readily available, and may have a lower risk of adverse events such as infections and traumatic catheter placement [33]. Options for single-use CIC catheters have become more prevalent, which may make them preferrable in public settings. However, single-use catheters have significant costs associated with them. In Ontario, Canada, the cost ranges between $1–2.30 each for a standard non-lubricated catheter, and ranges from $5–10 for a hydrophilic catheters [34]. In the United States, reimbursement for single-use CIC catheters ranges from $2–8 depending on the design and associated durable medical equipment reimbursement code. In [35] As the average patient catheterizes between 4–5 times a day, there can be a significant financial burden in the thousands of dollars for those choosing single-use catheters (especially if not fully covered by one’s insurance), something that may influence an individual’s bladder management decision. 

## 4. Discussion

Prior to the introduction of CIC in 1972, many patients with NLUTD (many of whom were allowed to leak into diapers as their primary bladder management) would prematurely die because of hydronephrosis, renal failure, and urinary infections. With an improved understanding of bladder physiology and the outcomes of improper bladder management, CIC is now considered the gold standard in bladder management in patients with NLUTD [36], and guidelines recommend CIC be performed in patients who are unable to effectively empty their bladder to prevent bladder overdistension [36].

Despite being considered the gold standard, however, not all individuals with NLUTD adopt and adhere to CIC in the long term. There are several important internal and external barriers that can impact the use of CIC in people living with NLUTD. With regard to internal barriers, the medical community has the potential to offer solutions. From a conservative perspective, high-quality teaching for female patients by a qualified nurse can help make the CIC experience less traumatic and maximize CIC adoption. In addition, a realistic assessment of hand function (with the involvement of an occupational therapist where necessary) can help avoid the patient’s frustration with being recommended CIC when in fact it may struggle for the patient to perform. Caregivers should be encouraged to understand CIC and how they can help in certain situations, but it is not realistic for many caregivers to take this on as daily requirement. Whether a caregiver is carrying out CIC or not, the use of overactive bladder medications (i.e., beta agonists and anticholinergic medications) or intravesical onabotulinum toxin can help increase the time between CIC occurrences, and is likely to decrease the daily time requirement for patients [37]. These medical therapies also serve to treat and reduce the risk of urinary incontinence, which is an important potential negative consequence of CIC. Anatomic differences or impaired UE motor function limitations can also be addressed in motivated individuals with the creation of a catheterizable channel; although, as mentioned above, significant time, effort, and morbidity must be accepted when going through these efforts. 

Many external barriers to performing CIC are specific to the design of public restrooms and the unique needs of NLUTD patients. While external barriers are more difficult for healthcare providers to influence, continued partnerships with SCI, MS, or SB advocacy groups can help promote the needs of people with NLUTD and influence regulations and public perception of CIC. Working to improve public restroom facilities, which can interfere with CIC in NLUTD patients either due to their limited number of handicapped stalls, limited accommodations within handicapped stalls, and the perceived cleanliness of the facility, may make CIC easier in public settings. Furthermore, education around UTI prevention, sharing tips and tricks for CIC in substandard situations, and patient acceptance in certain situations (for example, when a person is vacationing) of a short-term indwelling catheter or a prescription for self-start antibiotics in case of a UTI may be helpful in increasing patients’ bladder-specific quality of life. 

## 5. Conclusions

CIC, while considered the most effective of bladder management solutions in those with NLUTD who have bladder emptying problems, can be difficult for patients to adopt due to intrinsic and extrinsic barriers. A better understanding of these barriers is necessary to understand how to counsel patients and advocate for beneficial policy changes. Only after these barriers are addressed will improvements in quality of life and social participation be achievable. 

## Data Availability

Not applicable.

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
