# Peer review of "Internal and External Barriers to Bladder Management in Persons with Neurologic Disease Performing Intermittent Catheterization"

_ijerph, 2023, doi:10.3390/ijerph20126079_

Round 1
Reviewer 1 Report
Dear authors congratulations for your hard work. It is common knowledge that patients performing CISs (with or without help) are facing an everyday drama, having to deal with all the situations you have described.
If i may, I would like to share some comments upon your work.
1. What about the cost of the caregivers, besides family members?
2. What about the age of the patients? Older patieents might find it more difficult to perform CICs.
3. What about the trend of using reusable catheters?
Anyway, the issue that you brought up needs to be communicated and needs to be dealt, since we have an enormous number of our fellow citizens who unfortunately what you described is their reality, and we owe to our society to find ways to make it more bearable and easy.
Author Response
Reviewer 1
Dear authors congratulations for your hard work. It is common knowledge that patients performing CISs (with or without help) are facing an everyday drama, having to deal with all the situations you have described.
If i may, I would like to share some comments upon your work.
Comment 1: What about the cost of the caregivers, besides family members?
REPLY: We appreciate the reviewer’s perspective and agree. We have added a sentence to address patient costs if hiring outside of insurance.
Comment 2: What about the age of the patients? Older patients might find it more difficult to perform CICs.
REPLY: We thank the reviewer for this comment. We have included a section on the effects of age.
Comment 3: What about the trend of using reusable catheters?
REPLY: While this is a topic garnering attention in recent studies, we feel that this question, these trends, and how they affect CIC participation remains unclear and falls outside of the scope of this article.
Comment 4: Anyway, the issue that you brought up needs to be communicated and needs to be dealt, since we have an enormous number of our fellow citizens who unfortunately what you described is their reality, and we owe to our society to find ways to make it more bearable and easy.
REPLY: We thank the reviewer for this helpful comment and we fully agree.
Reviewer 2 Report
General comments:
- Review for errors in grammar and missing punctuation (ie commas and periods)
Abstract:
-Recommend rewording the sentence as follows:
"The use of catheters presents 12 unique challenges, based on both the person’s inherent characteristics on the external 13 limitations imposed by public toilets."
Introduction:
- Should say "to manage a patient's bladder" if referring to singular.
- Explain "NGB" populations
Internal Barriers:
- Recommend rewording the sentence as follows:
"Singly or in combination, these barriers can lead individuals to feel that CIC is a daunting long-term management option. (there should be a comma, missing period at the end)
Women:
- "In the either situation,..."
Lack of caregiver assistance:
- This sentence is not clear: "In these situations, it has been shown that adherence to a catheterization schedule is often not rigorous with patients who are reliant on their caregivers for catheterization often being reluctant to increase the number of catheterizations per day as they do not want to increase caregiver burden."
Urinary Incontinence
- "self-catharize"= self-catheterize
External barriers:
Appropriate space:
- Recommend rewording the sentence as follows: "In addition, CIC often requires multiple supplies positioned near the toilet (lubricant, urine drainage receptacle etc.) with clean shelves or tables on which to unpack and place the necessary equipment; however, these are usually not available."
Lack of facilities:
- "handicap assessable"= handicap accessible
- Many public facilities, however, are only partially compliant...
- "...friends’ homes to self-catheterize; nonetheless, this can be..."
Spaces in use:
- "In urgent urinary scenarios, those with NLUTD..." (missing comma)
Lack of public understand:
- "As a consequence, many in the general..." (missing comma)
Type and cost:
- Extra space between "lower risk of" and "adverse events"
- Error in the beginning of the sentence-- ". , options for single-use CIC..." = "Options for single-use CIC..."
Discussion:
- Error- "In addition, aA realistic assessment..."
- Error- "limited accommodations within handicapped stalls,, and..." (too commas)
- Error- "...prevention, sharing..." (missing a space after the comma)
There are quite a few errors in grammar, spelling, and punctuation. Some corrections listed in the Comments Section but recommend reviewing the paper closely for fix additional errors.
Author Response
Reviewer 2
Comment 1: General comments: Review for errors in grammar and missing punctuation (ie commas and periods)
REPLY: We have addressed the errors in grammar and punctuation. We appreciate the corrections.
Comment 2: Abstract:
-Recommend rewording the sentence as follows:
"The use of catheters presents 12 unique challenges, based on both the person’s inherent characteristics on the external 13 limitations imposed by public toilets."
REPLY: We appreciate the reviewer’s recommendation. We choose to not place numerical values on the internal and external challenges, as there are new challenges that will likely arise and we would be remiss to suggest that we have covered all of the possibilities. We changed the sentence to state “multiple unique challenges” to be more comprehensive and reflect the spirit of the reviewers concern.
Comment 3: Introduction:
- Should say "to manage a patient's bladder" if referring to singular.
- Explain "NGB" populations
REPLY: We have changed the introduction to reflect your recommendation “to manage a patient's bladder." We have clarified the last sentence by changing it to state “…because of literature demonstrating that it decreases overall morbidity in populations with NLUTD.” NGB was changed to NLUTD (which has previously been described – thanks for catching this error.
Comment 4: Internal Barriers:
- Recommend rewording the sentence as follows:
"Singly or in combination, these barriers can lead individuals to feel that CIC is a daunting long-term management option. (there should be a comma, missing period at the end)
REPLY: We have reworded to state, “These barriers can be daunting to individuals who select CIC as their long-term management option.”
Comment 5: Women:
- "In the either situation,..."
REPLY: Thank you. “The” has been removed.
Comment 6: Lack of caregiver assistance:
- This sentence is not clear: "In these situations, it has been shown that adherence to a catheterization schedule is often not rigorous with patients who are reliant on their caregivers for catheterization often being reluctant to increase the number of catheterizations per day as they do not want to increase caregiver burden."
REPLY: This is an important point made by the reviewer. It now reads “In these situations, it has been shown that adherence to a catheterizing schedule is often not rigorous, possibly because patients who are reliant on their caregivers for catheterization may be reluctant to increase the number of catheterizations per day out of fear of increasing caregiver burden.”
Comment 7: Urinary Incontinence
- "self-catharize"= self-catheterize
REPLY: We have corrected the spelling to reflect “self-catheterize.”
Comment 8: External barriers:
Appropriate space:
- Recommend rewording the sentence as follows: "In addition, CIC often requires multiple supplies positioned near the toilet (lubricant, urine drainage receptacle etc.) with clean shelves or tables on which to unpack and place the necessary equipment; however, these are usually not available."
REPLY: We thank the reviewer for this insightful comment. We have changed the sentence to reflect your recommendation.
Comment 9: Lack of facilities:
- "handicap assessable"= handicap accessible
- Many public facilities, however, are only partially compliant...
- "...friends’ homes to self-catheterize; nonetheless, this can be..."
REPLY: Thank you for your feedback. We have made the recommended changes.
Comment 10: Spaces in use:
- "In urgent urinary scenarios, those with NLUTD..." (missing comma)
REPLY: We have added the comma.
Comment 11: Lack of public understand:
- "As a consequence, many in the general..." (missing comma)
REPLY: Thank you. We have added the comma.
Comment 12: Type and cost:
- Extra space between "lower risk of" and "adverse events"
- Error in the beginning of the sentence-- ". , options for single-use CIC..." = "Options for single-use CIC..."
REPLY: Thank you. We have reviewed and edited the sentences.
Comment 13: Discussion:
- Error- "In addition, aA realistic assessment..."
- Error- "limited accommodations within handicapped stalls,, and..." (too commas)
- Error- "...prevention, sharing..." (missing a space after the comma)
REPLY: Thank you.
Reviewer 3 Report
I want to congratulate the authors for the review entitled " Internal and External Barriers to Bladder management in persons with neurologic disease performing intermittent catheterization", which gives a very good look over the existing problems and difficulties of persons with neurologic disease which perform CIC.
The scientific content is appropriate and well structured, my concerns are regarding minor spelling errors and punctuation:
-line 103 - reference should be noted at the end of the sentence
-line 113 "self-catharise"
-line 116 "altogether"
-line 134 "conducive"
-line 183 - use of punctuation between sentences
-line 198 "aA"
-line 217 "stalls,, "
-line 219 "prevention,sharing" punctuation
-line 227 please revise the last sentence
Also, I have concerns regarding Reference section, where I would recommend an intensive reviewing according to MDPI reference standards. Some references do not mention the year of publication (4, 5, 25,31) while others mention access date instead of publication date.
The article has minor spelling and text editing errors.
Author Response
Reviewer 3
Comment 1: I want to congratulate the authors for the review entitled " Internal and External Barriers to Bladder management in persons with neurologic disease performing intermittent catheterization", which gives a very good look over the existing problems and difficulties of persons with neurologic disease which perform CIC.
The scientific content is appropriate and well structured, my concerns are regarding minor spelling errors and punctuation:
-line 103 - reference should be noted at the end of the sentence
-line 113 "self-catharise"
-line 116 "altogether"
-line 134 "conducive"
-line 183 - use of punctuation between sentences
-line 198 "aA"
-line 217 "stalls,, "
-line 219 "prevention,sharing" punctuation
-line 227 please revise the last sentence
REPLY: We have addressed the spelling errors and punctuation. Thank you for helping us to correct these errors.
Comment 2: Also, I have concerns regarding Reference section, where I would recommend an intensive reviewing according to MDPI reference standards. Some references do not mention the year of publication (4, 5, 25,31) while others mention access date instead of publication date.
REPLY: Thank you for your feedback. References were made using Zotero. We have applied MDPI reference standards which uses the ACS Style Guide. For ‘websites and online materials’ ACS citation does not include year of publication, however it includes access date. Below is a link to a MDPI reference guide.
Reviewer 4 Report
An interesting work titeld “Internal and External Barriers to Bladder Management in Persons with Neurologic Disease Performing Intermittent Catheterization”,.
The manuscript addresses the important topic of barriers for people Performing Intermittent Catheterization due to neurogenic bladder. On the one hand, the work lacks scientific overtones, and on the other hand, it makes the medical staff dealing with the above-mentioned patients aware of how important this problem is for the interested parties themselves. 1. Discussion too short, more attention should be focused on medical indications for self-catheterization and describe how important this procedure is in the treatment of patients, e.g. after spinal cord injuries or with multiple sclerosis. 2. What does Aa mean in verse 198, is it a misspelling. 3. Abbreviations must be explained what OAB medications mean?
4. There is no reference to other studies on this subject in the discussion. In the discussion section, only one subject (32) is quoted.
Author Response
Reviewer 4
Comment 1: An interesting work titeld “Internal and External Barriers to Bladder Management in Persons with Neurologic Disease Performing Intermittent Catheterization”,.
The manuscript addresses the important topic of barriers for people Performing Intermittent Catheterization due to neurogenic bladder. On the one hand, the work lacks scientific overtones, and on the other hand, it makes the medical staff dealing with the above-mentioned patients aware of how important this problem is for the interested parties themselves.
REPLY: We thank the reviewer for this comment.
Comment 2: Discussion too short, more attention should be focused on medical indications for self-catheterization and describe how important this procedure is in the treatment of patients, e.g. after spinal cord injuries or with multiple sclerosis.
REPLY: We thank the reviewer for this feedback. We have included a new paragraph in the discussion. As to the Discussion being too short, this is a product of the fact that most of the “Discussion” is housed in other portions of this review.
Comment 3: What does Aa mean in verse 198, is it a misspelling.
REPLY: We have changed the word to state “a.”
Comment 4: Abbreviations must be explained what OAB medications mean?
REPLY: We have changed the sentence to state “overactive bladder medications (i.e. beta agonist and anticholinergic).”
Comment 5: There is no reference to other studies on this subject in the discussion. In the discussion section, only one subject (32) is quoted.
REPLY: Thank you for this comment. This is a product of the fact that most of the “Discussion” is housed in other portions of this review.
Reviewer 5 Report
Very well orgonized and presented manuscript, focused on the analysis of the internal and external Barriers to bladder management in persons with neurologic disease performing intermittent Catheterization.
I would like to mention the following in order to further improve the quality of your work:
A seperate section should be included, which would present the criteria for the collection of all the relevant articles that are referred to your manuscript. This means that you could mention the keywords and the databases you have utilized in order to support your valuable article.
A seperate and more detailed conclusion section would be valuable.
Author Response
Reviewer 5
Comment 1: Very well orgonized and presented manuscript, focused on the analysis of the internal and external Barriers to bladder management in persons with neurologic disease performing intermittent Catheterization.
I would like to mention the following in order to further improve the quality of your work:
A separate section should be included, which would present the criteria for the collection of all the relevant articles that are referred to your manuscript. This means that you could mention the keywords and the databases you have utilized in order to support your valuable article.
REPLY: We thank the reviewer for this interesting point. We used PubMed to search literature within the last 25 years based on the following keywords: spinal cord injury, upper extremity motor function, urinary incontinence, caregivers and barriers, with additional papers included that are known to the authors. For the purposes of this paper we did not perform a formal systematic review as it was beyond the scope of this invited communication/opinion piece.
Comment 2: A seperate and more detailed conclusion section would be valuable.
REPLY: Thank you, we have specified a conclusion section.
Round 2
Reviewer 4 Report
I have no more commentsAuthor Response
It does not appear that the reviewer requested any modifications.